# Trends of Lipophilic, Antioxidant and Hematological Parameters Associated with Conventional and Electronic Smoking Habits in Middle-Age Romanians

**DOI:** 10.3390/jcm8050665

**Published:** 2019-05-12

**Authors:** Mihaela Badea, Laura Gaman, Corina Delia, Anca Ilea, Florin Leașu, Luis Alberto Henríquez-Hernández, Octavio P. Luzardo, Mariana Rădoi, Liliana Rogozea

**Affiliations:** 1Faculty of Medicine, Transilvania University of Brasov, 500019 Brasov, Romania; mihaela.badea@unitbv.ro (M.B.); ancamariailea@yahoo.com (A.I.); mradoi_unitbv@yahoo.com (M.R.); r_liliana@yahoo.com (L.R.); 2“Carol Davila” University of Medicine and Pharmacy, 050474 Bucharest, Romania; glauraelena@yahoo.com; 3National Institute for Mother and Child Health “Alessandrescu-Rusescu”, 20395 Bucharest, Romania; corina_delia@yahoo.com; 4Toxicology Unit, Clinical Sciences Department, Universidad de Las Palmas de Gran Canaria, Paseo Blas Cabrera Felipe, s/n, 35019 Las Palmas de Gran Canaria, Spain; luis.henriquez@ulpgc.es (L.A.H.-H.); octavio.perez@ulpgc.es (O.P.L.)

**Keywords:** conventional smokers, electronic cigarettes users, plasma lipophilic components, oxidative stress

## Abstract

It is known that cigarette smoking is correlated with medical associated inquires. New electronic cigarettes are intensively advertised as an alternative to conventional smoking, but only a few studies demonstrate their harmful potential. A cross-sectional study was designed using 150 subjects from Brasov (Romania), divided into three groups: non-smokers (NS = 58), conventional cigarettes smokers (CS = 58) and electronic cigarettes users (ECS = 34). The aim of this study was to determine levels of some plasma lipophilic and hematological components, and the total antioxidant status that could be associated with the smoking status of the subjects. Serum low density lipoproteins (LDL) cholesterol increased significantly for ECS participants versus NS group (18.9% difference) (*p* < 0.05). Also, the CS group is characterized by an increase of serum LDL cholesterol (7.9% difference vs. NS), but with no significant statistical difference. The variation of median values of serum very low density lipoproteins (VLDL) was in order NS < ECS < CS, with statistical difference between NS and CS groups (34.6% difference; *p* = 0.023). When comparing the antioxidant status of the three groups, significant differences (*p* < 0.05) were obtained between NS vs. CS and NS vs. ECS. Similar behavior was identified for CS and ECS. Statistically significant changes (*p* < 0.0001) for both vitamin A and vitamin E were identified in the blood of NS vs. CS and NS vs. ECS, and also when comparing vitamin A in the blood of the CS group versus the ECS group (*p* < 0.05). When all groups were compared, the difference in the white blood cell (WBC) was (*p* = 0.008). A slight increase in the red blood cell (RBC) count was observed, but with no statistical difference between groups. These results indicated that conventional cigarette and e-cigarette usage promotes the production of excess reactive oxygen species, involving different pathways, different antioxidants and bioactive molecules.

## 1. Introduction

Smoking is an important issue to deal with for scientists from different countries. Considering the recent reports of the World Health Organization (WHO) that indicate the highest prevalence of tobacco smoking among adults (28%) and some of the highest prevalence of tobacco use by adolescents. According to the Eurobarometer 2017, conducted by the European Commission in Romania, tobacco consumption remains relatively high, with a smoking prevalence of 28% among those aged over 15, similar to the European average (26%). The top five countries with the highest prevalence were Greece (37%), France (36%), Bulgaria (36%), Croatia (35%) and Latvia (32%). The lowest prevalence was reported in Sweden (7%). The reported prevalence in Romania is 38% among men and 19% among women [1,2].

Smoking is inherited as a risk factor in developing different pathological conditions because tobacco smoke contains high concentrations of oxidants and reactive oxygen species (ROS). Thus, smoking can be considered as the main cause of a decrease in the production capacity of antioxidant systems due to the increased production of ROS [3,4,5].

Recently, observing that tobacco use has fallen, the industry has introduced an alternative known as electronic cigarettes (e-cigarettes), which are said to be healthier and would be an alternative to smoking. Since then, the number of electronic cigarette users has increased significantly due to the perception that they would be a healthier substitute for tobacco use and lower or non-existent risk [6]. Therefore, electronic cigarettes have become the most used smoking products, and unfortunately young people pick up smoking habits with these products that are considered less dangerous [7].

Although it was initially claimed that electronic cigarettes are free of harmful effects, it has become increasingly known that they have a detrimental effect on health. There is increasing evidence that electronic cigarettes emit toxic substances such as nicotine, organic volatile substances, carbonyls and other airborne particles that affect both smokers (active smokers) and non-smokers (secondhand and third hand smokers) [8,9,10].

An increasing number of in vitro and in vivo studies demonstrate a range of adverse effects of both the vapors obtained from e-cigarettes as well as the nicotine-containing fluid. Vapors of e-cigarettes have adverse effects on both cultured cells and living animals. Various outcomes have been measured in models. Electronic cigarettes induce inflammation, augment the development of allergic airway inflammation in asthma models, change the behavior of animals, and suppress pulmonary host defense [11].

Tobacco smoke contains high concentrations of oxidants and reactive oxygen species, present both in the gaseous state and tar state. It has been estimated that each smoke puff contains about 1016 oxidants [12] Cigarettes contain many chemical compounds, including nicotine, carbon monoxide, nitrogen, and free radicals such as hydroxyl, superoxide, and peroxide [12]. These compounds are associated with an increased incidence of cancer (lung, mouth, renal, etc.) [7,11,12].

The most important enzymatic antioxidant systems reported are superoxide dismutase (SOD), glutathione peroxidase (GPx), glutathione reductase (GR) and catalase (CAT). A decrease was observed in the level of superoxide dismutase (SOD) activity from healthy non-smokers to light smokers to heavy smokers [13]. There were studies where increased CAT, SOD and GR activities were observed with high lipid peroxidation levels especially in smoking and alcohol-smoking groups [14].

Also, there were studies that demonstrated the deep involvement of non-enzymatic antioxidant systems in the fight with ROS species generated during oxidative stress associated with tobacco consumption.

Glutathione (GSH) protects cellular components against the effects of hydrogen peroxide and other hydroperoxides by providing reduction equivalents [15]. Long-term exposure to cigarette smoke causes a reduction in GSH levels. Chronic cigarette smoking evokes a lung glutathione (GSH) adaptive response that results in elevated GSH levels in the lung epithelial lining fluid [16].

Air pollution, heavy metals and other endocrine-disrupting chemicals (EDCs) and smoking habits influence the concentration of serum vitamin D [17]. Smokers constantly overexposed to free radicals contained in tobacco that cause depletion of plasma and tissue deposits of micronutrients such as vitamins A, E, and C. There are studies that have shown that tobacco constituents reduced the levels of several vitamins of the B-complex [18].

Active and passive tobacco cigarette smoking increased white blood cell, lymphocyte, and granulocyte counts for at least one hour in smokers and non-smokers (*p* < 0.05) [19]. Smoking is associated with changes of lipoproteins profiles as decreased high-density lipoprotein (HDL) cholesterol and elevated triglycerides [20].

Recent studies performed by our group [21] indicated the association of different inorganic elements (such as rare earth elements (REE) and heavy metals) with the smoking status (conventional cigarette smokers versus electronic cigarettes users). We have found that smoking is mainly a source of heavy metals while the use of e-cigarettes is a potential source of REE. However, these elements were detected at low concentrations, probably also because of the middle age of the participants to our study (mainly young people).

The aim of this study was to correlate the biochemical values of the blood parameters as lipid components (lipoproteins groups, total cholesterol, triglycerides) and other possible biomolecules involved in oxidative stress (uric acid, fat-soluble vitamins) with smoking status (conventional and electronic).

## 2. Study Design and Participants Analysis

### 2.1. Study Design

We conducted a cross-sectional study that included 150 Romanian subjects. The recruitment was thought for a determined period time in a prospective way. All the subjects responded to a call made to participate in the present investigation. Recruitment was performed between December 2017 and February 2018. The series was formed by middle age Romanians, who considered themselves “healthy”—58 non-smokers (NS), 58 conventional cigarette smokers (CS), and 34 e-cigarette users (ECS). All users of e-cigarette were ex-smokers. However, dual users—defined as persons who smoke cigarettes and use e-cigarette at the same time—were excluded from the study. Smoking addiction was classified according to the number of the cigarette or heets/day—group A (1–9 cigarettes /day), group B (10–14 cigarettes/day), and group C (more than 15 cigarettes/day), and respectively group eA (1–9 heets/day), group eB (10–14 heets/day), and group eC (more than 15 heets/day). The classification was based on the self-reports of the participants.

Demographical data was obtained and a face-to-face interview, aimed to know details about the smoking status and life style, was also done. Data was recorded on paper and subsequently digitalized for statistical analysis. Participation in the study was totally free and no one received any compensation.

### 2.2. Ethical Statement

All participants signed an informed consent before taking the sample. The study design was approved by the Ethical Committee of the Faculty of Medicine, Transilvania University of Brasov, Romania. The present study was conducted in accordance with the guidelines of Transilvania University Ethical Commission (Approval 2017) and international rules [22,23].

### 2.3. Blood Samples Analysis

Blood samples were collected from all of the participants. All samples were taken in the morning and the participants were asked not to smoke or use e-cigarette prior to the blood collection. Samples of blood were collected in 4 mL K3-EDTA tube with vacuum for hematology (ENGLOBER VAC), and in tubes with separator gel (Becton-Dickinson Vacutainer^®^ serum separation tubes). The serum and plasma were separated within a maximum of 2 h after collection. The collected samples were labeled and coded and kept at −20 °C until biochemical analysis.

Using venous blood drawn in tubes with K3 EDTA as anticoagulant, a hemoleucogram was done with a fluorescence flow cytometry analyzer using LASER semiconductor and hydrodynamic focusing [24].

Concentrations of total cholesterol, high density lipoprotein (HDL), triacylgliceride (TG) were analysed through enzymatic colorimetric methods [25,26] with limit of detection (LOD) 3 mg/dL, 3 mg/dL and 4 mg/dL respectively. Using Cobas Integra 400 Plus (Roche Diagnostics Ltd., Rotkreuz, Switzerland) was used to determine the levels of low density lipoprotein (LDL) cholesterol (enzymatic colorimetric assay-bicromatic endpoint at 583 nm), albumin (colorimetric assay with endpoint method; LOD 2 g/L), uric acid (enzymatic colorimetric method—Cobas Integra Uric Acid Cassette; Roche Diagnostics) [27,28].

Vitamin A/E HPLC Kits (ImmunDiagnostik Method—UHPLC, certificate IVD) were used to determine plasma vitamin A and E, with Agilent Technologies—UHPLC 1290 Infinity II (Agilent Technologies, Inc, Waldbronn, Germany), where LOD were 0.05 mg/L for vitamin A and 1 mg/L for vitamin E [29,30,31]. Serum antioxidant status was determined by an EDEL system biosensor (EDEL meter) (Edel Therapeutics, Lausanne, Switzerland), an analytical device, which includes a biological detector coupled with a chemical transducer and specific software) [32].

### 2.4. Statistical Analysis

Descriptive analyses were conducted for all variables. Arithmetic means, standard deviation (SD), medians, ranges and percentiles 25th and 75th of the distribution were calculated for continuous variables. Proportions were calculated for categorical variables.

The normality of the data was tested using the Kolmogorov Smirnov test. Most of the data (i.e., concentrations of compounds) did not follow a normal distribution. As a consequence, we chose not assume a normal distribution in any case, and comparisons between groups were performed using a non-parametric test (Kruskal–Wallis or Mann–Whitney *U* test).

Differences in the categorical variables were tested by the chi-squared test. We used PASW Statistics version 19.0 (SPSS Inc., Chicago, IL, USA) and GraphPad Prism version 8.0 (GraphPad Software, San Diego, CA, USA) to manage the database of the study and to perform statistical analyses. Probability levels of *p* < 0.05 (two tailed) were considered statistically significant.

## 3. Results

The results are divided into the general data coming from interview concerning their principal characteristics and characteristics of blood biochemical (lipids, vitamins, ions) and hematological parameters.

### 3.1. Characterisation of Study Volunteers

Subject characteristics are provided in Table 1.

A total of 150 subjects were included in the present study, distributed in three different groups as follows: non-smokers (*n* = 58), cigarette smokers (*n* = 58), and e-cigarette users (*n* = 34). The groups contained more females than males, but the structures of the groups were similar, without statistically differences between groups.

In order to understand the behavior of smokers, the participants were asked about their reasons for taking up smoking, previous attempts to quit smoking and the withdrawal symptoms associated with recommencing smoking (Table 2.).

### 3.2. Characterisation of Blood Components

The biochemical data concerning hydrophobic components (total cholesterol, lipoprotein fractions, triglycerides, fat-soluble vitamins), albumins, uric acid and total antioxidant status of the three groups of the subjects (Appendix A) were statistically analyzed (arithmetic mean ± STDEV, median and corresponding percentiles p25th–p75th) (Table 3).

In order to have a better overview of the significantly statistic differences between groups, a rainbow chart was obtained (Figure 1).

Statistical analysis (arithmetic mean ± STDEV, median and corresponding percentiles p25th–p75th) for biochemical detection was carried out (Appendix A) taking into consideration possible gender differences and also the influence of smoking addiction. Rainbow charts were constructed for correlations between biochemical compounds detected and p values corresponding to smoking status differences between male and female groups (Figure 2) and between subgroups of cigarette and e-device users (Figure 3).

Flow cytometry is one of the most powerful tools for single-cell analysis with different applications in diagnosis and progression evaluation in immunology [33], hematology, and blood cancers [34,35]. Using this method, characteristics of blood parameters for all the groups are obtained and a synthesis of their data is indicated in Table 4 and the statistical approaches are presented in Figure 4.

## 4. Discussion

Smoking was associated with increased risks of many chronic diseases that shorten life and decrease quality of life [36]. The main objectives of the present study aim to investigate their smoking habits and some associated biological effects.

Previous studies have indicated that the average age for a first-time smoker is 14.5 years, and the average daily smoker is 17.7 years [37]. Approximately 20% of children in the final-year of studies are smokers. Early onset of tobacco use contributes to the high rate of addiction, making adolescence a vulnerable age, with various mental pathologies of children and adolescents [38,39].

Even though recently there has been identified an increase in young and middle age people who use e-devices for using nicotine/tobacco [40,41], the general number of users is not as high as for conventional smokers and non-smokers, so the final number of volunteers from this study using e-cigarettes was a little bit lower than the other two, but similar or even higher than in other studies [42,43]. A similarity was observed with our study where our Romanian participants reported 17.3 ± 3.9 years for CS groups and 18.4 ± 6.1 years for ECS group (Table 1). Tobacco data survey from 31 December 2016 [44] indicated a high prevalence trend of tobacco use in young Romanian people (ages around 15) as 20.0% for male and 17.0% for female, which need to be an element of concern in order to propose and to continue better plans for smoking cessation for younger generation.

It can be seen that biggest reason declared to start smoking is the entourage (family and friends) (77.6% for CS and 76.5% for ECS group). Work/school pressure is the second biggest reason reported by the participants (Table 2). Other reasons for smoking were indicated as relaxing (4 persons CS), curiosity (2 persons CS), pleasure (1 person CS and 1 person ECS), good smell/flavor (1 ECS), stress due to divorce, attempting to lose weight, pampering, distressing, etc. Our results are in line with data reported recently for Italian people which indicated as top reasons for smoking: the influence of friends (61.1%), the enjoyment and satisfaction (15.6%), to feel mature and independent (9.0%), because of the influence of partner/family (6.6%), because of stress (2.5%), to feel more secure (1.9%) and for the sake of curiosity (1.8%) [45].

In one the survey, conducted at a private university preparatory training center on the Anatolian side of Istanbul, students declared curiosity was the most important reason to start smoking (39%), and for the second row they indicated the encouragement of tobacco user friends (30.6%) [46].

Adverse effects due to smoking habits (Table 2) reported by 44.8% people from the CS group involved in our study as: dizziness (53.9%), nausea (23.1%), upset stomach (15.4%), diarrhea (7.7%) and others—sore throat (15.4%); cough (15.4%); cough accompanied by sputum (7.7%); face acne (7.7%). Also, four people from ECS group reported adverse effects: one person reported a combination of dizziness/unclear view/nausea, two people reported only dizziness and one person reported losing weight.

Probably knowing the harmful effects of cigarettes (conventional especially), some of the smokers reported a tentativeness to quit smoking or at least to reduce the number of smoking episode/day. Reasons to quit were associated for three (25.0%) women from ECS group during their pregnancy period. But we have to underline that 13.8% from CS and 26.47% from ECS have not attempted to quit smoking. Analyzing the data reported in Table 2, it can be observed that some of the participants reported some symptoms associated with smoking cessation, considering them as reason to restart smoking: weakness (6.1%—for CS and 16.7%—for ECS), trembling hands (6.1%—for CS and 8.3%—for ECS), and tachycardia (3.0%—for CS). However, it can be observed again the influence of the entourage convinced people to restart smoking (100% reported by ECS and 45.5% reported by CS) and the temptation of “just a cigarette” (60.6%—for CS and 50.0%—for ECS).

Other symptoms associated with smoking cessation were reported in the literature: chest tightness, concentration problems, constipation, coughing/clearing your throat/dry throat/postnasal drip, irritability or anxiety, depression, hunger [47].

The volunteers enrolled in our study declared that they don’t have any specific disease and they consider themselves as healthy people. Considering the age of the participants (Table 1), all the findings could be suggestive for middle age population. The participants were asked not to smoke before having blood drawn, but studies have indicated that tobacco and e-cigarette smoking exposure does not acutely alter the response of the antioxidant system, neither under active nor passive smoking conditions. It was demonstrated that total antioxidant capacity, catalase activity (CAT) and reduced glutathione (GSH) remained similar to baseline levels immediately after and 1 h-post exposure of persons who smoked two cigarettes within 30 min and a group of active e-cigarette smoking persons who smoked a pre-determined number of puffs in the same time, using a liquid containing 11 ng/mL nicotine [48].

Cigarette smoking is associated with changes in the blood lipid profile that can be incriminating to the pathway of atherosclerosis development [49].

In a longitudinal study, smokers who quit had significantly increased their HDL cholesterol levels after one year [50]. Smoking children and adolescents (8–19 years old) involved in different studies have serum LDL cholesterol and triglyceride significantly higher than the non-smokers group and lower concentration of serum HDL cholesterol [51,52].

In a previous meta-analysis of 54 epidemiologic studies, cigarette smokers, compared with nonsmokers, were found to have significantly higher serum concentrations of cholesterol (3% difference), triglycerides (9% difference), and very low density lipoprotein (VLDL) cholesterol (10% difference), and a decrease with 6% HDL cholesterol level than nonsmokers [53].

Similar findings of changes of lipid profile induced by smoking were also observed in our study. The level of total cholesterol increased for cigarette smoking people (4.7% difference from the median value of nonsmokers), but the higher median value was observed for e-cigarette users (8.8% difference from the median value of nonsmokers). No statistical significance was obtained between our groups, related to their gender. Considering the statistical analysis presented in Figure 3, a statistically difference was observed when all the subgroups A, B and C of conventional cigarettes smokers were analyzed versus NS (*p* = 0.037). Analyzing the *p* values of each subgroups of addicted conventional smokers vs. NS groups, it has been observed that the higher number of cigarettes indicated a higher difference between them, and that the higher significance was obtained when the subgroup C was compared with NS group (*p* = 0.017). Scientists consider that acrolein, a foul-smelling vapor that is produced by burning tobacco, could be associated with cardiovascular disease risk in humans [54]. Acrolein is easily absorbed into the bloodstream through the lungs and it contributes to heart disease by affecting the way the body metabolizes cholesterol [55], by associating with lipid peroxidation triggered by oxidative stress [56] and it has been detected in human atherosclerotic lesions [57].

Due to their role in reverse cholesterol transport from peripheral tissues to the liver for excretion, high density lipoproteins (HDL) are considered athero-protective, by promotion of endothelial homeostasis and inhibition of monocyte adhesion [58]. Under oxidative stress, such as metal ions [59], endogenous oxidants [60], and as well as environmental factors and unhealthy lifestyle (poor diet and tobacco use) [61], HDL is susceptible to modification that may decrease or eliminate HDL’s athero-protective effects, leading to a “dysfunctional” particle. Similar to other mentioned studies, a slight decrease was observed in HDL cholesterol in the serum of CS (with 1.7% lower) and ECS (with 4.6% lower) respectively, compared with the NS group.

Smoking has a negative impact on the amount and function of HDL, which may explain the increased risk of cardiovascular disease in smokers [62]. There was a higher concentration of HDL cholesterol in former smokers than current smokers [20,63]. The mechanisms by which smoking decreases HDL cholesterol are incompletely understood. Smoking is involved in the increase of catecholamine releasing, the increase of VLDL and LDL concentrations and the reduction of HDL-C concentrations due to a possible surge in free fatty acids circulation [64].

In our study (Appendix A, Figure 1, Table 3), serum LDL cholesterol, named the “bad cholesterol”, increased significantly for ECS participants versus the NS group (18.9% difference) (*p* < 0.05). Also the CS group is characterized by an increase of serum LDL cholesterol (7.9% difference vs. NS), but with no significant statistical difference. Considering the results from Appendix A, Figure 2, no significant differences were observed when gender distribution was analysed. An increase was observed in the difference of LDL cholesterol levels between the e-user subgroups vs. NS according to the number of heets/day, but without any significantly difference.

The variation of median values of serum VLDL was in order NS < ECS < CS, with statistical difference between NS and CS groups (34.6% difference; *p* = 0.023). It seems that, for our participants, e-cigarette consumption affects VLDL concentration (26.9% vs. NS) less than conventional cigarette smoking. The difference between the values of VLDL in serum of subgroups of smokers and e-users subgroups increased when the number of cigarettes/day or heets/day increased. No significant statistical differences (*p* > 0.05) were obtained for CS subgroups when analysed. But it has to be mentioned that the statistical significance (*p* = 0.049) was obtained when eA vs. eB vs. eC vs. NS subgroups were analysed and a high significance was obtained especially when eC and NS were compared (*p* = 0.009).

In humans, uric acid is the most abundant aqueous antioxidant, accounting for up to 60% of the serum free radical capture capacity, and is an important agent for eliminating intracellular free radicals during metabolic stress, including smoking [65,66]. High plasma concentrations of uric acid may provide protection in situations characterized by an increase in cardiovascular risk and oxidative stress, such as smoking. There are studies showing a decrease in uric acid plasma levels in smokers [67] and the reduction of antioxidants, including uric acid, in smokers, indicating that oxidative stress increases each time a cigarette is smoked. Other studies have shown that uric acid administration increases circulating antioxidant protection and allows the restoration of endothelium-dependent vasodilation [68,69].

We have to note that the uric acid seems to be an important marker of smoking habits also for our groups also. A statistical difference (*p* < 0.0001) was obtained for serum uric acid if we compared NS and CS groups and a statistical significance (*p* < 0.001) was also obtained if NS and ECS were compared. No significantly statistical differences were obtained between CS and ECS groups. It has to be noticed that *p* < 0.05 were obtained when the men in groups ECS vs. NS and respectivelly CS vs. ECS were analysed (Appendix A, Figure 2). For the women in the groups (F) significant differences were obtained for CS vs. ECS vs. NS (*p* < 0.05) and for ECS vs. NS (*p* < 0.01).

In a study involving 300 volunteers [68], in opposition to our findings, plasma uric acid concentration was significantly lower in smokers than in nonsmokers and a statistically significant negative correlation was noted between the smoking status parameters, including both the number of cigarettes smoked/day. The explanation for this variation could be the age differences between the two studies: the average age for the non-smoking group was 38.0 ± 17.5 versus 24.5 ± 6.7 in our study and the average age for the cigarette smoking group was 35.6 ±16.0 versus 28.4 ± 10.8 in our study, gender distribution (higher number of men in their study, and higher number of women in our study) and the number of participants (higher in their situation). One of the most important differences was in the number of cigarette per day: in their study they also had heavy smokers with more than 20 cigarettes per day, and in our study all the reported values were at a maximum 20 cigarettes per day, the calculated average value being 11.0 ± 5.9 cigarettes per day.

Very interesting findings were obtained when nicotine addiction was considered (Appendix A, Figure 3) and the subgroups were analysed. Uric acid values depend on the number of cigarettes and heets per day reported by our volunteers, with *p* = 0.015 when A vs. B vs. C vs. NS, and *p* = 0.004 when eA vs. eB vs. eC vs. NS were compared. Increasing uric acid levels in diseases associated with increased oxidative stress may still be seen as an adaptive defense mechanism against radical attack, the possibility that uric acid may function as a pro-oxidant cannot be excluded. Perhaps the anti- or prooxidant effects of uric acid manifest in one sense or the other according to the specific conditions. By controlling the excretion of uric acid, the kidney is in the position of an important organ for maintaining the antioxidant capacity of the body [70].

Antioxidant activity of albumin may also be attributed not only to the concentration of albumin in plasma, but also to its ability to modify its structure and to link transition metals, thereby diminishing their availability to participate in redox reactions [71].

The role of albumin as a scavenger of free radicals is already well defined [72]. Antioxidant activity of albumin is attributed to the free-thiol group of cysteine 34 as a free group [73]. Albumin is the main (but not the only) source of free thiol groups in the serum, but it is an important redox buffer, being considered a sensitive indicator of oxidative stress in the vascular compartment [74] and the main scavenger of electrophiles of the blood [75]. Relatively recent research shows albumin plays an active role in the defense of plasma antioxidants by its peroxidase activity, and this plasma protein cannot be considered as just a sacrificial antioxidant [76].

Using the Edel system [32] it was possible to determine the antioxidant status of the three groups. Significant differences (*p* < 0.05) were obtained when NS vs. CS and NS vs. ECS were compared. Similar behavior was identified for CS and ECS. The total antioxidant status was measured in the other study [77] and it was found significantly higher in the non-smoker group than in the smoker group, similarly to our results.

Analyzing the differences between vitamin A and vitamin E in these three groups (Appendix A, Figure 1, Table 3), we found these vitamins as possible early markers to characterize the changes of antioxidant status of these groups, despite the low numbers of cigarettes/heets used per day. Significantly statistical changes (*p* < 0.0001) were identified for both vitamin A and vitamin E in the blood of the NS vs. CS and NS vs. ECS. We have to mention that a significant statistical difference was obtained corresponding to vitamin A when comparing to CS group versus ECS group (*p* < 0.05).

When the levels of vitamin A of female groups were compared, *p* < 0.0001 was obtained for all three groups’ comparison, as well as for CS vs. NS and ECS vs. NS. Also, in CS vs. ECS, a significant difference (*p* = 0.009) in the level of vitamin A in females was obtained. For men subgroups the situation is a little bit different because *p* < 0.05 was obtained for all three groups when compared, as well as for CS vs. NS and ECS vs. NS, but no significant difference was obtained when CS vs. ECS were compared. The values of vitamin A in the blood of C vs. eC subgroups are different and *p* = 0.057. Vitamin E acts as a chain-breaking antioxidant involved as scavenger of peroxyl radicals and in limitation of the propagation of lipid peroxidation. There are in vitro studies that led to the hypothesis that cigarette smokers require additional dietary vitamin E and vitamin C [78]. Authors investigated the effects of active and passive smoking on the activity of antioxidant enzymes and antioxidant micronutrients and they found a significantly lower mean of serum vitamin E level in cigarette smokers compared to nonsmokers [79], probably due to the impaired absorption and bioavailability of vitamin E in smokers. More so, tobacco smoke contains numerous compounds, many of which are oxidants and prooxidants, capable of producing free radical and enhancing the oxidative stress [80,81]. In “smoking” rats, lung oxidation product of vitamin E (quinone structure) increased, but the level of plasma vitamin E was unchanged [82]. The regulatory mechanisms that control plasma concentration of vitamin E concentrations are largely unknown. Studies of vitamin E metabolism in normal subjects have demonstrated that the total plasma vitamin E pool is replaced daily [83]. Also, it was obtained that smokers had lower lymphocyte and platelet α-tocopherol concentrations; no differences were observed between groups in plasma α- and γ-tocopherol concentrations [84]. In our studies, for the samples tested and analysed (for NS—three missing values and eight values lower than 10 ng/mL; for CS—one missing value and 13 values lower than 10 ng/mL; for ECS—9 missing values and three values lower than 10 ng/mL), it was observed that the level of vitamin E increased significantly (*p* < 0.0001) in CS and ECS groups versus NS group. hen vitamin E concentration for all subgroups (A, B, C, eA, eB and eC) was compared with the NS group (Appendix A), statistically significant differences were obtained, excepting for eA vs. NS (Figure 3). It seems that a small number of heets/day do not have a significant influence on the level of vitamin E and that vitamin A could be considered as an earlier biomarker of smoking (the level of vitamin A varied significantly even for eA vs. NS; *p* = 0.003). Supplementary studies are necessary in order to more accurately define the important role of vitamin E in daily diet, particularly under oxidative stresses, such as cigarette smoking [85].

Cigarette smoking (number of cigarettes smoked per day or the depth of inhalation) is associated with higher leukocyte concentrations, suggesting that smoking induces a sustained, long-term inflammatory response [36,86]. Previous studies show that a high leukocyte count is associated consistently with the intensity of cigarette smoking and also the effect persists also after quitting [87].

For our both groups of smokers (traditional cigarette and e-cigarette) white blood cell (WBC) counts were elevated versus nonsmoking group, in accord with other epidemiological data indicating that smoking is associated with elevated WBC [86,88,89]. The *P* value obtained with the Kruskal-Wallis test for NS vs. Cs vs. ECS indicated a statistically significant difference (*p* = 0.008) (Figure 2). Mann Whitney test performed for NS vs. CS groups indicated a statistically significant difference (*p* = 0.002). The atherogenic effect of cigarette smoking may be partially mediated by leukocytes and WBC counts could be considered as useful biomarker of endothelium damage. A slight increase of red blood cell (RBC) count was observed, but with no statistic difference between groups.

Similar to our study, earlier studies on blood monocyte counts in smokers reported an increased monocyte blood count compared to non-smokers [89,90,91,92], while no significant changes in the monocyte percentage were identified [82]. Significant changes in monocytes were observed for our groups also for NS vs. CS (*p* = 0.001) as well as for NS vs. ECS (*p* = 0.05). Analysis between our three groups yielded statistical differences.

There were also observed some low increasing counts of basophiles and eosinophils for both our smoking groups, comparing with nonsmokers, but not statistical differences were obtained. The eosinophil granulocytes have been associated with respiratory diseases [93,94,95,96,97]. Similar levels of blood eosinophils in smokers and non-smokers have been found in some studies [97,98,99,100,101].

Hematocrit and hemoglobin levels are elevated in our group of smokers (CS and ECS), because of elevated red cell volume [102]. The elevation of hemoglobin concentration is explained as compensatory mechanism of decreased oxygen delivering capacity due to the exposure to carbon monoxide in smoker’s blood [102,103,104,105]. The Mann Whitney test indicated a significantly statistical difference for hematocrits (*p* = 0.03) for NS vs. CS groups. No significant differences between the groups were observed for hemoglobin values.

Nicotine addiction is the second leading cause of death worldwide and the leading cause of premature death. The efficacy of nicotine replacement/treatment in younger people is unknown [106].

While most studies show a positive relationship between e-cigarette consumption and smoking cessation, the evidence remains inconclusive due to the low quality of research published so far. Performing well-controlled longitudinal randomized studies on the population is needed to elucidate the role of electronic cigarettes in smoking cessation.

Firm warnings need to be associated with extensive knowledge and telephone help lines in order to change the attitude towards smoking. A smoke-free policy should be a priority for global smoke control [107].

### Strengths and Limitations of the Study

One of the major limitations of this study is attributed to the grouping of the participants. Due to the design of the study, based on self-reports of the subjects about smoking, the misclassification of nonsmokers/smokers/e-cigarette users must be considered as a possible bias. Possible biases of this study have to be taken into account: (i) some differences in age distribution (that can influence the levels of certain blood components) or (ii) majority of women (whose serum levels of certain components may be affected by gender). In addition, a potential selection bias can derive from the voluntary nature of the participants, most of the participants considering themselves to be “healthy” despite being heavy smokers or heavy e-cigarette users.

Other variables associated to the diet/food supplements/medications intake/alcohol consumptions, and other lifestyle issues (sport, secondary/tertiary smoking) have to be taken further into account, or even the cigarette brand or the e-cigarette model (containing information about voltage and battery power) and concentration of nicotine/product, because one-year follow-up data was not yet coded for analysis.

Notwithstanding these limitations, we consider that this work has important strengths. First, this type of young and middle age Romanian population is rarely assessed at all, and most of these bioactive blood components have been rarely determined and compared in a study involved both conventional and electronic smoking possibilities. Although the populations were limited, for many components a statistical difference was found between non-smokers, smokers of cigarettes or users of e-cigarettes, or even for two of these groups. Given the cross-sectional nature of the study design there is no possibility of establishing causality. Even so, these results, despite their limitations, are highly interesting and encourage further research, as well as address some valuable conclusions to people involved in health promotion and stakeholders involved in smoking reduction/cessation.

## 5. Conclusions

The study design proposed an investigation on how tobacco smoke exposure could modify some lipophilic and haematological parameters and on the efficacy of antioxidant defenses. Taking into account the results of our study based on the self-reported questionnaire and also based on scientific data obtained from biochemical and hematological analyses, we concluded that conventional and electronic cigarette smoking may influence oxidative stress by affecting the levels of plasma antioxidants and other biological active compounds, which may be involved in the mechanisms underlying various diseases. As this reduction is proportionate to the smoking status and leads to cardiovascular diseases, it is recommended to stop or reduce smoking.

In particular, young people should be protected by banning all forms of tobacco (conventional and new models of nicotine intake) advertising and promotion. The work contains much useful data about the metabolic consequences of smoking and e-smoking in young people. Changes in smoking prevalence in younger age groups should be monitored. Educating young people on tobacco dependence and its effects on health should remain an important part of education. The young people themselves need to be involved in examining constructive ways to reduce smoking initiation [108,109].

## Figures and Tables

**Figure 1 jcm-08-00665-f001:**
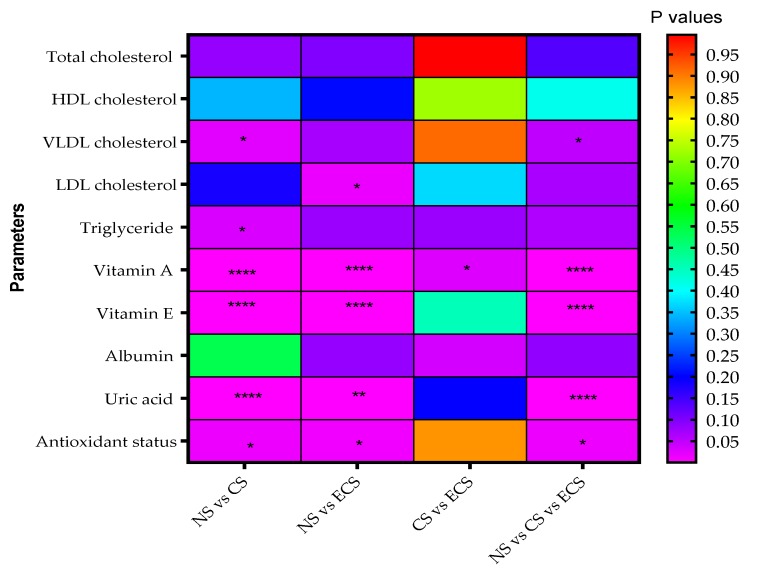
Rainbow chart for correlations between biochemical compounds detected and *p* values corresponding to smoking status differences between groups. The color close to violet—smaller *p* value (more significant statistical difference); the color close to red—high *p* value (low difference between groups); NS—Non-smokers; CS—Cigarette smokers; ECS—Electronic cigarette users; * *p* < 0.05; ** *p* < 0.01; **** *p* < 0.0001.

**Figure 2 jcm-08-00665-f002:**
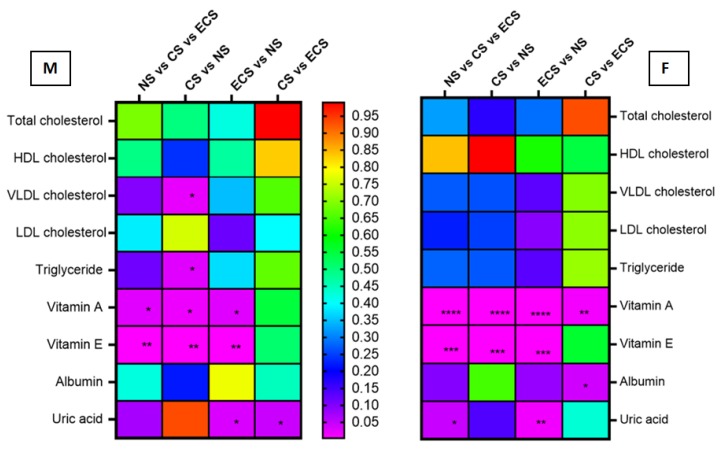
Rainbow chart for correlations between biochemical compounds detected and *p* values corresponding to smoking status differences between gender subgroups of cigarettes and e-devices users. The color close to violet—smaller *p* value (more significant statistical difference); the color close to red—high *p* value (low difference between groups); NS—Non-smokers; CS—Cigarette smokers; ECS—Electronic cigarette users; M-male; F-female; * *p* < 0.05; ** *p* < 0.01; *** *p* < 0.001; **** *p* < 0.0001.

**Figure 3 jcm-08-00665-f003:**
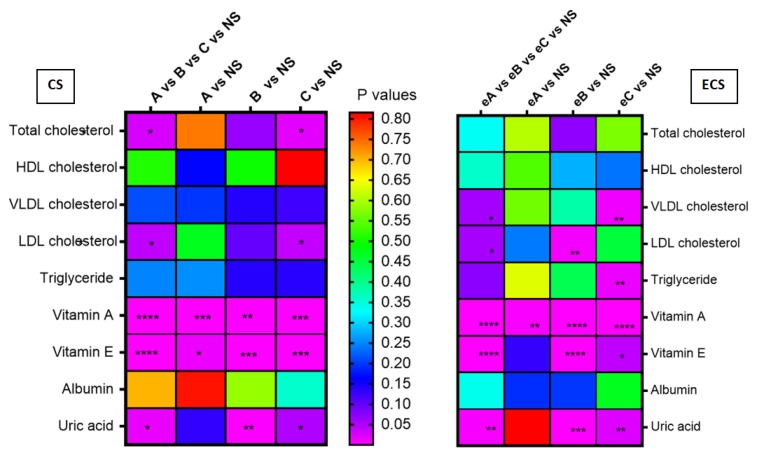
Rainbow chart for correlations between biochemical compounds detected and *p* values corresponding to smoking status differences between subgroups. The color close to violet—smaller *p* value (more significant statistical difference); the color close to red—high *p* value (low difference between groups); NS—Non-smokers; CS—Cigarette smokers; ECS—Electronic cigarette users; group A (1–9 cigarettes /day), group B (10–14 cigarettes/day), and group C (more than 15 cigarettes/day), and respectively group eA (1–9 heets /day), group eB (10–14 heets/day), and group eC (more than 15 heets/day); * *p* < 0.05; ** *p* < 0.01; *** *p* < 0.001; **** *p* < 0.0001.

**Figure 4 jcm-08-00665-f004:**
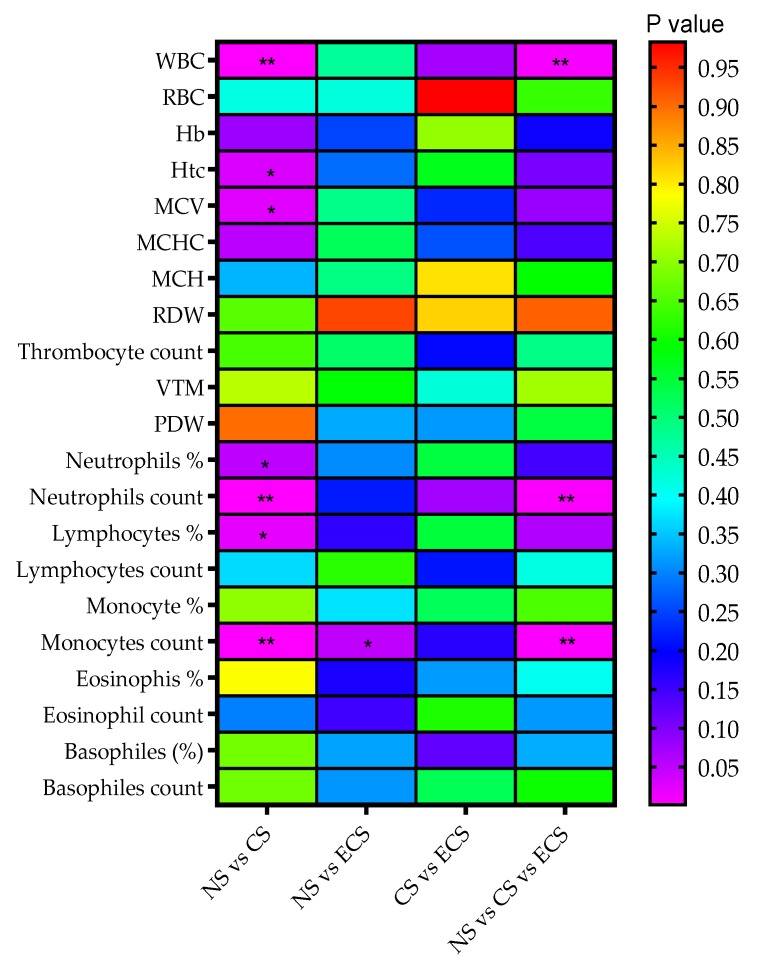
Rainbow chart for correlations between hematologic data detected in group samples and *p* values corresponding to smoking status differences between groups. Where NS—Non-smokers; CS—Cigarette smokers; ECS—Electronic cigarette users; WBC—white blood (leucocytes); RBC—red blood cell (erythrocyte); Hb—hemoglobin, Htc—hematocrit; MCV—mean corpuscular volume, MCHC—mean corpuscular hemoglobin concentration; MCH—mean erythrocyte hemoglobin; RDW—red cell distribution width; VTM—mean of platelet volume; PDW—platelet distribution width. The color close to violet—smaller *p* value (more significant statistical difference); the color close to red—high *p* value (low difference between groups); * *p* < 0.05; ** *p* < 0.01.

**Table 1 jcm-08-00665-t001:** General characteristics of the subjects involved in the study.

Characteristics	Non-Smokers (*n* = 58)		Cigarette Smokers (*n* = 58)		E-Cigarette Users (*n* = 34)		*P* Value
	*N*	%	*N*	%	*N*	%	
Gender							0.307
Male	10	17.2	17	29.3	8	23.5	
Female	48	82.8	41	70.7	26	76.5	
Habitat							0.012 ^(a)^
Urban	39	67.2	45	77.6	32	94.1	
Rural	19	32.8	13	22.4	2	5.9	
BMI (kg/m^2^) ^(b)^							0.078
<18.5	8	14.0	3	5.3	1	2.9	
18.5–24.99	39	68.4	35	61.4	19	55.9	
25–29.99	8	14.0	14	24.6	8	23.5	
>30	2	3.5	5	8.8	6	17.6	
Age (years) (mean ± STDEV)	24.5 ± 6.7		28.4 ± 10.8		35.2 ± 9.4		<0.001 ^(c)^
Age start smoking (mean ± STDEV) (years)	-		17.3 ± 3.9		18.4 ± 6.1		0.002 ^(d)^
Time smoking cigarettes (mean ± STDEV) (years)	-		10.5 ± 5.9		14.3 ± 8.0 *		
Cigarettes/day (mean ± STDEV)	-		11.0 ± 5.9		13.0 ± 5.7 *		
Time using e-cigarettes (mean ± STDEV) (months)					16.2 ± 15.5		n.a.

STDEV—standard deviation; BMI—body mass index; n.a.—not applicable; *—smoking in the past. ^(a)^—Chi square test; ^(b)^—2 missed values; ^(c)^—Kruskal-Wallis test; ^(d)^—Mann-Whitney *U* test.

**Table 2 jcm-08-00665-t002:** Smokers answers concerning reasons for start/re-starting smoking and reported adverse effects.

Item	Suggested Answer	Cigarette Smokers (*n* = 58)		E-cigarette Users (*n* = 34)	
		*N*	%	*N*	%
Reasons of start smoking	Social entourage	45	77.59%	26	76.47%
	Partner pressure	1	1.72%	0	0.00%
	Children pressure	0	0.00%	0	0.00%
	Work pressure	10	17.24%	3	8.82%
	Religious	0	0.00%	0	0.00%
	Esthetic	1	1.72%	0	0.00%
	Existent disease	0	0.00%	0	0.00%
	Other reasons	12	20.69%	6	17.65%
Previous attempts to quit smoking	No	8	13.79%	9	26.47%
	Just a decreasing of frequency	17	29.31%	13	38.24%
	Yes	33	56.90%	12	35.29%
Re-starting smoking/withdrawal symptoms	Trembling hands	2	6.06%	1	8.33%
	Tongue tremors	0	0.00%	0	0.00%
	Eye tremors	0	0.00%	0	0.00%
	Nausea and vomiting	0	0.00%	0	0.00%
	Weakness	2	6.06%	2	16.67%
	Tachycardia	1	3.03%	0	0.00%
	Sweating	0	0.00%	0	0.00%
	Entourage	16	45.45%	12	100.00%
	"Just a cigarette"	20	60.61%	6	50.00%
	Other reasons	8	24.24%	7	58.33%
Reported adverse effects	No	28	48.28%	28	82.35%
	No answer	4	6.90%	2	5.88%
	Yes	26	44.83%	4	11.76%
Examples of adverse effects	Constipation	0	0.00%	0	0.00%
	Diarrhea	2	7.69%	0	0.00%
	Dizziness	14	53.85%	3	75.00%
	Unclear view	0	0.00%	1	25.00%
	Upset stomach	4	15.38%	0	0.00%
	Insomnia	0	0.00%	0	0.00%
	Sleepiness	1	3.85%	0	0.00%
	Nausea	6	23.08%	2	50.00%
	Others	12	46.15%	1	25.00%

**Table 3 jcm-08-00665-t003:** Quantitative levels of hydrophobic components (total cholesterol, lipoprotein fractions, triglycerides, fat-soluble vitamins), albumins, uric acid and total antioxidant status in serum of non-smokers, cigarette smokers and e-cigarette users.

Parameter	Non-Smokers (*n* = 58)		Cigarette Smokers (*n* = 58)		E-Cigarette Users (*n* = 34)	
	Mean ± STDEV	Median (p25th–p75th)	Mean ± STDEV	Median (p25th–p75th)	Mean ± STDEV	Median (p25th–p75th)
Total cholesterol (mg/dL)	164.29 ± 33.68	160.00 (138.75–185)	179.64 ± 47.84	167.50 (146.75–207.75)	173.24 ± 27.19	174.00 (156.50–189.75)
HDL cholesterol (mg/dL)	58.72 ± 12.28	56.95 (50.55–66.55)	58.22 ± 18.01	56.00 (47.18–66.78)	55.29 ± 13.66	54.35 (45.65–65.98)
VLDL cholesterol (mg/dL)	15.59 ± 6.84	13 (11.00–18.00)	23.19 ± 25.33	17.50 (11.75–24.25)	20.15 ±12.40	16.50 (12.00–25.25)
LDL cholesterol (mg/dL)	100.16 ± 32.10	95 (74.00–117.50)	111.00 ± 41.88	102.50 (85.75–132.75)	112.03 ± 26.67	113 (94.50–134.50) ^(a)^
Triglycerides (mg/dL)	78.22 ± 34.07	66.50 (56.00 –89.25)	115.78 ± 126.63	87.00 (57.50–122.00)	100.91 ± 61.95	83.50 (60.50–125.25)
Vitamin A (mg/L)	0.39 ± 0.10	0.37 (0.32–0.46)	0.52 ± 0.14	0.49 (0.42–0.62)	0.59 ± 0.14	0.56 (0.49–0.67) ^(a)^
Vitamin E (mg/L)	10.35 ± 3.12	9.93 (8.27 –11.91)	13.98 ± 6.25	12.91 (10.69–15.5)	14.07 ± 4.20	13.29 (10.65–16.88) ^(a)^
Albumin (g/dL)	4.86 ± 0.25	4.85 (4.65–5.00)	4.84 ± 0.27	4.82 (4.67–4.97)	4.95 ± 0.33	4.94 (4.79–5.12) ^(a)^
Uric acid (mg/dL)	4.36 ± 0.97	4.30 (3.70–5.00)	4.85 ± 1.28	4.7 (3.7–5.7)	5.24 ± 1.43	5.2 (4.15–6.2) ^(a)^
Antioxidant status (Edel units/s)	−1.15 ± 0.32	−1.112 (−1.81–−0.72)	−1.29 ± 0.31	−1.252 (−1.838–−0.76)	−1.28 ± 0.24	−1.27 (−1.72–−0.88)

NS—Non-smokers (*n* = 58); CS—Cigarette smokers (*n* = 58); ECS—Electronic cigarette users (*n* = 34), ^(a)^—one missing value (number of ECS = 33).

**Table 4 jcm-08-00665-t004:** Hematological parameters of subjects.

Parameter	Non-Smokers (*n* = 58)		Cigarette Smokers (*n* = 58)		E-Cigarette Users (*n* = 34)	
	Mean ± STDEV	Median (p25th–p75th)	Mean ± STDEV	Median (p25th–p75th)	Mean ± STDEV	Median (p25th–p75th)
WBC count (10^3^μL)	6.47 ± 1.52	6.37 (5.41–7.39)	7.42 ± 1.87	7.39 (6.23–8.75)	6.74 ± 1.63	6.6 (5.39–8.13)
RBC count (10^3^/μL)	4.74 ± 0.39	4.69 (4.43–4.91)	4.82 ± 0.49	4.73 (4.42–5.13)	4.8 ± 0.36	4.71 (4.56–5.04)
Hb (g/dL)	13.69 ± 1.47	13.70 (12.85–14.33)	14.25 ± 1.47	14 (13.2–15.3)	13.94 ± 1.71	13.95 (13.1–14.93)
Htc (%)	40.97 ± 3.41	40.85 (39.00–42.73)	42.53 ± 3.72	42.20 (39.83–44.53)	41.71 ± 4.14	41.75 (39.28–44.55)
MCV (fL)	86.47 ± 3.83	86.50 (84.98–89.40)	88.37 ± 4.35	88.6 (85.4–90.45)	86.8 ± 5.7	87.00 (85.35–89.63)
MCHC (pg/cell)	28.88 ± 2.11	29.10 (28.08–30.10)	29.58 ± 1.75	29.7 (28.7–30.43)	29 ± 2.67	29.05 (28.40–30.25)
MCH (g/dL)	33.38 ± 1.48	33.20 (32.60–34.30)	33.46 ± 1.00	33.35 (32.8–34.15)	33.34 ± 1.27	33.30 (32.80–34.13)
RDW (%)	13.07 ± 1.43	12.70 (12.38–13.20)	12.81 ± 0.79	12.70 (12.20–13.40)	13.00 ± 1.33	12.70 (12.38–13.05)
Thrombocyte count (10^3^/μL)	270.97 ± 75.40	255.00 (217.50–308.75)	258.93 ± 48.27	254.00 (224.00–293.25)	269.35 ± 47.45	267.50 (234.00–297.25)
VTM (fL)	10.60 ± 0.99	10.40 (9.80–11.20)	10.50 ± 0.85	10.35 (9.80–11.23)	10.61 ± 0.79	10.50 (10.18–11.30)
PDW (fL)	12.65 ± 2.23	12.00 (11.18–13.80)	12.49 ± 1.75	12.15 (11.20–13.33)	12.76 ± 1.66	12.55 (11.75–13.85)
Neutrophils (%)	54.19 ± 8.57	53.65 (48.98–58.53)	57.30 ± 8.14	55.85 (52.00–64.35)	56.16 ± 8.31	54.85 (50.55–61.03)
Neutrophils count (10^3^/μL)	3.55 ± 1.23	3.45 (2.60–4.04)	4.33 ± 1.50	4.21 (3.39–5.14)	3.82 ± 1.20	3.69 (2.93–4.44)
Lymphocytes (%)	34.56 ± 7.93	35.05 (29.43–39.33)	31.42 ± 6.85	31.95 (25.75–36.83)	32.46 ± 7.64	32.9 (27.23–36.88)
Lymphocytes count (10^3^/μL)	2.20 ± 0.61	2.12 (1.71–2.64)	2.28 ± 0.58	2.36 (1.84–2.60)	2.16 ± 0.68	1.96 (1.70–2.57)
Monocytes (%)	8.10 ± 1.85	7.90 (6.60–9.33)	8.16 ± 1.56	7.95 (6.88–9.33)	8.35 ± 1.56	8.30 (7.28–9.40
Monocytes count (10^3^/μL)	0.52 ± 0.18	0.49 (0.41–0.57)	0.59 ± 0.14	0.59 (0.48–0.68)	0.55 ± 0.13	0.56 (0.47–0.61)
Eosinophil (%)	2.57 ± 3.43	1.85 (1.00–2.83)	2.56 ± 2.63	1.90 (1.28–2.65)	2.41 ± 1.23	2.05 (1.58–3.10)
Eosinophil count (10^3^/μL)	0.17 ± 0.22	0.12 (0.06–0.18)	0.18 ± 0.19	0.14 (0.08–0.21)	0.16 ± 0.09	0.15 (0.11–0.24)
Basophiles (%)	0.58 ± 0.27	0.50 (0.40–0.70)	0.55 ± 0.26	0.50 (0.40–0.70)	0.62 ± 0.25	0.60 (0.40–0.83)
Basophiles (10^3^/μL)	0.04 ± 0.01	0.04 (0.03–0.05)	0.04 ± 0.02	0.04 (0.03–0.04)	0.04 ± 0.02	0.04 (0.03–0.06)

WBC—white blood (leucocytes); RBC—red blood cell (erythrocyte); Hb—hemoglobin, Htc—hematocrit; MCV—mean corpuscular volume, MCHC—mean corpuscular hemoglobin concentration; MCH—mean erythrocyte hemoglobin; RDW—red cell distribution width; VTM—mean of platelet volume; PDW—platelet distribution width.

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
