# Peer review of "Trends of Lipophilic, Antioxidant and Hematological Parameters Associated with Conventional and Electronic Smoking Habits in Middle-Age Romanians"

_jcm, 2019, doi:10.3390/jcm8050665_

Round 1
Reviewer 1 Report
The work is interesting despite that the group with e-cig is smaller than the other two. There are many information about studied subjects but it is not known how long the group with e-cig used to smoke normal cigarettes before they started e-smoking (they all were ex-smokers). Did they use similar type of e-cig, and do the liquid in the e-cigs was with nicotine (most probably yes but it is not wrriten). The data on C-reactive protein mentioned in Table 3 and Fig. 1 are missing. In Discussion, line 362 should be thiol group and not "free hydroxyl group of cysteine".
The work contain many useful data about the metabolic consequences of e-smoking in young people. The study should be continue.
Author Response
We are very thankful to the Editor/Reviewers for their notes; we have carefully read the comments and have revised / completed the manuscript accordingly. Our responses are given in a point-by-point manner below, as well all the changes to the manuscript are highlighted
Reviewers/Editor comments:
Reviewer #1
The work is interesting despite that the group with e-cig is smaller than the other two.
At the time we closed the recruitment, the number of e-cig users was smaller than the other two groups, but the importance and significance of the study was underlined also using this group. We added a paragraph in the paper
Even that in the last period it was identified an increase of young and middle age population which use e-devices for using nicotine/tobacco (Alexander et al, 2019; Simmons et al., 2016), the general number of users is not so high as for conventional smokers and no smokers, so the final number of volunteers from this study using e-cigarettes was a little bit lower than the other two, but similar or even higher than in other studies (Dawkins et al, 2018; Avino et al., 2018; ).
There are many information about studied subjects but it is not known how long the group with e-cig used to smoke normal cigarettes before they started e-smoking (they all were ex-smokers).
Thank you for your suggestion. The table 1 was updated with data concerning smoking in the past of conventional cigarettes by the group of e-users: 14.3 ± 8.0 years for time of smoking cigarettes and number of cigarettes/day - 13.0 ± 5.7 (marked in red in the manuscript) and also was mentioned in the body text.
Did they use similar type of e-cig, and do the liquid in the e-cigs was with nicotine (most probably yes but it is not written).
The e-cigarettes users declared different types of brands of products, with different concentration of nicotine/product. This was noted supplementary as limitation of the study.
The data on C-reactive protein mentioned in Table 3 and Fig. 1 are missing.
Thank you for bringing this to our attention. We removed data from tables/fig and we forgot to remove it also from the text. Changes were done in the manuscript
In Discussion, line 362 should be thiol group and not "free hydroxyl group of cysteine".
Thank you for your observation. It was replaced “hydroxyl” with “thiol”
The work contain many useful data about the metabolic consequences of e-smoking in young people.
Thank you
The study should be continue.
Yes!

Reviewer 2 Report
This paper describes a cross-sectional study where 150 subjects voluntarily participate to a call which aims to investigate their smoking habits and some associated biological effects.
The study design is interesting since the researchers have proposed an investigation on how tobacco smoke exposure could modified some lipophilic and haematological parameters and on the antioxidant defences efficacy. These purposes have been achieved, focusing on possible differences among conventional and electronic smoking addiction. However, in my opinion, several methodological concerns have to be mentioned:
- The sample size estimation is not provided, and it is not clear how the calculation was made: on the basis of which outcome? what is the expected between-group difference? what is the literature reference?
- Inclusion criteria and how the data were collected are unclear: do the authors used a standardized questionnaire? how the smoke addiction/exposure is classified? do the author evaluate the possible sex-related differences?
- the statistical analyses are very simple and do not evaluate some possible confounding factors that could influence the antioxidant levels, such as diet or general health status.
In general, the article is vague and it is not clear what are the main objectives of the project. Finally, the English language needs an extensive editing and revision.
Author Response
Dear Ms. Ivana Zhang
Assistant Editor
Journal of Clinical Medicine
We are very thankful to the Editor/Reviewers for their notes; we have carefully read the comments and have revised / completed the manuscript accordingly. Our responses are given in a point-by-point manner below, as well all the changes to the manuscript are highlighted in blue.
Reviewers/Editor comments:
Reviewer 2
This paper describes a cross-sectional study where 150 subjects voluntarily participate to a call which aims to investigate their smoking habits and some associated biological effects.
The study design is interesting since the researchers have proposed an investigation on how tobacco smoke exposure could modified some lipophilic and haematological parameters and on the antioxidant defences efficacy.
Thank you
These purposes have been achieved, focusing on possible differences among conventional and electronic smoking addiction.
Thank you for your support
However, in my opinion, several methodological concerns have to be mentioned:
- The sample size estimation is not provided, and it is not clear how the calculation was made: on the basis of which outcome?
Thank you for bringing this to our attention. We mentioned now in the text a supplementary phrase in STUDY DESIGN - The recruitment was thought for a determined period time in a prospective way.
what is the expected between-group difference? what is the literature reference?
It was mentioned a supplementary phase to explain the differences between study groups from this article and also the similarity with other scientific literatures (new references were introduced in the text)
Even that in the last period it was identified an increase of young and middle age population which use e-devices for using nicotine/tobacco (Alexander et al, 2019; Simmons et al., 2016), the general number of users is not so high as for conventional smokers and no smokers, so the final number of volunteers from this study using e-cigarettes was a little bit lower than the other two, but similar or even higher than in other studies (Dawkins et al, 2018; Avino et al., 2018; ).
40. Alexander, J.P.; Williams, P.; Lee, Y.O.; Youth who use e-cigarettes regularly: A qualitative study of behavior, attitudes, and familial norms, Prev Med Rep. 2019; 13: 93–97.
41. Simmons, V.N; Quinn G.P.; Harrell, P.T.; Meltzer, L.R,; Correa, J.B.; Unrod, M.; Brandon, T.H. E-cigarette use in adults: a qualitative study of users' perceptions and future use intentions, Addict Res Theory. 2016;24(4):313-321.
42. Dawkins, L.; Cox, S.; Goniewicz, M.; McRobbie, H.; Kimber, C.; Doig, M.; Kosmider, L. ‘Real world’ compensatory behaviour with low nicotine concentration e-liquid: subjective effects and nicotine, acrolein and formaldehyde exposure. Addiction, 2018; 113(10):1874-1882. doi: 10.1111/add.14271
43. Avino P., Scungio M., Stabile L., Cortellessa G., Buonanno G., Manigrasso M. (2018). Second-hand aerosol from tobacco and electronic cigarettes: evaluation of the smoker emission rates and doses and lung cancer risk of passive smokers and vapers. The Science of the Total Environment, 9; 642: 137-147
- Inclusion criteria and how the data were collected are unclear: do the authors used a standardized questionnaire?
Inclusion criteria –new details mentioned supplementary in the STUDY DESIGN– middle age Romanians, who considered them “healthy”, as part of three groups: non-smokers (NS), conventional smokers (CS) and e-cigarette users (ECS)
The questionnaire was made ad hoc for the present study (it was not a standardized questionnaire). The full questionnaire included different data concerning: diet (Food Frequency Questionnaire-FFQ); hydration (Beverage Intake Questionnaire); physical activity (IPAQ); addiction to nicotine (Fagerstrom-Nadjari Test and WHO – Global Adult Tobacco Survey adaptations); lifestyle (adapted from FP7 project PlantLIBRA). It was mentioned as study limitation that other correlation were not done yet because 1-year follow-up data were not yet coded for analysis.
Data collection – mentioned in text as face to face interview – on paper-created data base
Yes, it was a face to face interview with the volunteers involved in the study; data were recorded on paper and subsequently digitalized for statistical analysis. The information is mentioned in the STUDY DESIGN.
how the smoke addiction/exposure is classified?
Thank you for your idea, that produced more and interesting data concerning the groups. Following your valuable point, it was classified the smoke addiction related to the number of the cigarette or heets/day- group A (1-9 cigarettes /day), group B (10-14 cigarettes/day), and group C (more than 15 cigarettes/day), and respectively group eA (1-9 heets /day), group eB (10-14 heets/day), and group eC (more than 15 heets/day).
And also supplementary data were analysed using these groups versus NS. The most important results are now part of this paper, too – supplementary file 3a and 3b; Figure 3 and comments in the DISCUSSION
do the author evaluate the possible sex-related differences?
Thank you for your idea. The groups were divided by gender and the comparative results are now mentioned in this paper – supplementary file 2a and 2b; Figure 2 and comments in the DISCUSSION
- the statistical analyses are very simple and do not evaluate some possible confounding factors that could influence the antioxidant levels, such as diet or general health status.
The full questionnaire included different data concerning: diet (Food Frequency Questionnaire FFQ); hydration (Beverage Intake Questionnaire); physical activity (IPAQ); addiction to nicotine (Fagerstrom-Nadjari Test and WHO – Global Adult Tobacco Survey adaptations); lifestyle (adapted from FP7 project PlantLIBRA). It was mentioned as study limitation that other correlation were not done yet because 1-year follow-up data were not yet coded for analysis.
In general, the article is vague and it is not clear what are the main objectives of the project.
There were summarized the main results and the sentences concerning the study and its main objectives/results were included now in RESULTS, at the beginning of the DISCUSSION, in CONCLUSION and in the ABSTRACT.
Finally, the English language needs an extensive editing and revision.
In this new form corrections were done using a specialized partner for English language supervision
We hope that all the updates and new introduced data, based on the valuable recommendation of the reviewers, would be considered, and the new form of our article will be considered suitable for publication.
Thank you in advance for your consideration
The Authors

Round 2
Reviewer 2 Report
The authors deeply revised the manuscript, taking into account all the
suggestions and advises proposed. The methodological and statistical
changes and the language revision deserve a special mention.